# Mechanical Determinants of Sprinting and Change of Direction in Elite Female Field Hockey Players

**DOI:** 10.3390/s23187663

**Published:** 2023-09-05

**Authors:** Alejandro Bustamante-Garrido, Mikel Izquierdo, Bianca Miarka, Ariel Cuartero-Navarrete, Jorge Pérez-Contreras, Esteban Aedo-Muñoz, Hugo Cerda-Kohler

**Affiliations:** 1Escuela de Ciencias del Deporte y la Actividad Física, Facultad de Salud, Universidad Santo Tomás, Santiago 8370003, Chile; abustamante7@santotomas.cl (A.B.-G.); joperezc@gmail.com (J.P.-C.); 2Navarrabiomed, Hospitalario Universitario de Navarra (HUN), Universidad Pública de Navarra (UPNA), IdiSNA, 31008 Pamplona, Spain; mikel.izquierdo@unavarra.es; 3Departamento de Educación Física, Deportes y Recreación, Facultad de Artes y Educación Física, Universidad Metropolitana de Ciencias de la Educación, Santiago 7760197, Chile; 4Laboratorio de Biomecánica Deportiva, Unidad de Ciencias Aplicadas al Deporte, Instituto Nacional de Deportes, Ministerio del Deporte de Chile, Santiago 7750000, Chile; 5Laboratory of Psychophysiology and Performance in Sports and Combats, Postgraduate Program in Physical Education, School of Physical Education and Sport, Federal University of Rio de Janeiro, Rio de Janeiro 21941-599, Brazil; miarkasport@hotmail.com; 6Federación Chilena de Hockey Sobre Césped, Santiago 7750000, Chile; arielcuartero87@gmail.com; 7Escuela de Doctorado, Universidad de las Palmas de Gran Canaria (EDULPGC), 35001 Las Palmas, Spain; 8Escuela de Ciencias de la Actividad Física, el Deporte y la Salud, Facultad de Ciencias Médicas, Universidad de Santiago de Chile, Santiago 9170022, Chile; 9Unidad de Fisiología del Ejercicio, Centro de Innovación, Clínica MEDS, Santiago 7691236, Chile

**Keywords:** physical and physiological analysis, training and game monitoring, motion analysis

## Abstract

Profile determination in field hockey is critical to determining athletes’ physical strengths and weaknesses, and is key in planning, programming, and monitoring training. This study pursued two primary objectives: (i) to provide descriptive data on sprinting, deceleration, and change of direction (COD) abilities and (ii) to elucidate the mechanical variables that influence sprint and COD performance in elite female field hockey players. Using radar and time-gate technology, we assessed performance and mechanical data from 30 m sprinting, deceleration, and COD tests for 26 elite female hockey players. A machine learning approach identified mechanical variables related to sprint and COD performance. Our findings offer a framework for athlete categorization and the design of performance-enhancing training strategies at the international level. Two pivotal mechanical variables—relative maximum horizontal force (F0) and maximum velocity (Vmax)—predominantly influence the times across all tested distances. However, the force–velocity profile (FVP) and horizontal deceleration do not influence the variance in the COD test outcomes. These insights can guide the design, adjustment, and monitoring of training programs, assisting coaches in decision making to optimize performance and mitigate injury risks for female hockey players.

## 1. Introduction

Decision making is essential in any sport [1], particularly in dynamic team sports such as field hockey, volleyball, soccer, rugby, and basketball. In this process, coaches and staff play a key role in planning, programming, and monitoring training [2]. In order to succeed, it is crucial to know the sport’s profile, including biomechanical, physiological, and technical-tactical factors. Therefore, profile determination is critical to determining athletes’ physical strengths and weaknesses and fundamental to making decisions about the training process [3].

Field hockey is an invasive team sport featuring many offensive and defensive abilities mixed with intermittent high-intensity running [2,4], which represent significant percentages of the entire game (12 to 26%) [5]. This profile is characterized by movements such as high-velocity running, jumping, changes of direction, kicks, and hits, demonstrating the power generated by the athlete [6]. Having high-intensity running capacity and maintaining these actions consistently in a game or season is an essential aspect of team sports [4] and can mean victory [5]. Similarly, changing direction (COD) is crucial, enabling competitors to evade opponents and gain advantageous positions [7,8]. COD is an implicit skill in agility, including acceleration, deceleration, and decision-making situations [9]. Accordingly, several aspects of sprinting and COD must be considered in field hockey athletes’ training process, such as duration, intensity, time spent at different intensities, and physical and mechanical determinants.

Regarding sprinting, the average duration in elite matches is 1.8 ± 0.4 s (s), and the longest sprints last 4.1 ± 2.1 s [10]. Additionally, sprints typically cover distances ranging from 10 to 20 meters (m) [11] and account for 1.4–2.1% (73–112 s) of the total game time [12]. Sprints are also performed at between 18 and 24 km/h, representing 91 ± 4% of the players’ maximum speed [12,13]. The applied horizontal force is one of the most relevant variables when executing a sprint [14,15,16]. The force–velocity profile (FVP) method allows the identification of mechanical variables such as the maximum horizontal force (F0), maximum velocity (V0), and maximum horizontal power (Pmax) applied during the sprint [17]. This method is a valuable tool for coaches and physical trainers in the determination of the physical profile of the athlete. Several studies have studied the profile of different sports, such as soccer [15,18], athletics [19], ice hockey [20,21], basketball, and tennis [22]. However, evidence of the force–velocity profile and the mechanical determinants of the sprint in elite female field hockey is lacking.

The ability to perform quick sprints and efficiently change direction is crucial for sport performance. On average, male players perform 1148 ± 128.9 change of direction movements during elite field hockey matches [23]. Furthermore, another study revealed 512 ± 69 acceleration–deceleration actions in elite female players [24]. Strength, power, speed, and technical ability determine the athlete’s ability to change direction and their acceleration–deceleration actions [9]. In addition, fast decelerations are crucial for quick change of direction (COD) movements performed during team sports [25,26]. However, the associations between mechanical factors, such as FVP and deacceleration, and COD in elite female field hockey has not been investigated.

The identification of the mechanical variables of sprinting (at different distances) and COD allows the identification of the athlete’s profile to optimize performance and determine risk factors for injury [27] and post-injury monitoring [28]. This study has two objectives: to present sprint, change of direction, and deceleration reference values, and to determine the influences of mechanical variables of the FVP and deceleration on sprint acceleration performance (at 5, 15, and 30 m) and the ability to change direction in elite female field hockey athletes. We hypothesized that sprint mechanical variables and deceleration significantly explain sprint and COD performance in elite female field hockey players.

## 2. Materials and Methods

### 2.1. Participants

Twenty-eight female players in the Chilean national team participated in this study. Table 1 shows participants’ characteristics (field hockey training experience, 18.9 ± 4.7 years; time representing the Chilean national team, 6.2 ± 4.9 years). Athletes voluntarily agreed to participate in this study, signing an informed consent form. The assessments were carried out following the Declaration of Helsinki standards. The tests were conducted in a competitive period, one week before a series of official world-ranking matches.

### 2.2. Design

The testing was conducted between 07:30 a.m. and 09:00 a.m. in Chile’s official field hockey arena. Before starting, the protocol of each evaluation was detailed to the participants after each athlete performed a general warm-up similar to the one they performed during their physical preparation sessions. The warm-up lasted 15 min, including low-moderate intensity cardiovascular exercises, mobility, and short-distance sprints. After the standardized warm-up, subjects performed two maximal sprints on an indoor track. They all performed a linear run with a maximum acceleration of 30 m to determine the force–velocity profile (FVP). This profile represents the force–velocity and power–velocity relationship that the neuromuscular system of the lower extremities is capable of generating [29]. Only the best times were considered for data analysis for the two trials.

#### 2.2.1. Speed-Acceleration Evaluation 30 m

We evaluated the maximum acceleration in 30 m on a field hockey field. For this, the Stalker ATS II model radar (Applied Concepts, Dallas, TX, USA; accuracy + 1.61 km/h, sampling 46.9 Hz) was located on a tripod 10 m from the starting line and at the height of 1 m to align with the location of the center of mass (CM) of the subject [30]. Radar technology relies on using sonic waves to ascertain an object’s distance. Additionally, velocity can be precisely measured through the Doppler effect (variation in wave frequency as an object approaches or recedes) [31]. The radar was operated from a computer to avoid the variability produced by manual operation [32]. Athletes were instructed to initiate with no backward step, performing two 30 m maximal sprints from a standing staggered-stance start, with at least 5 min passive recovery between sprints. Sprint performance (split times 0–5, 0–15, and 0–30 m) and mechanical outputs were computed for the best time trial.

#### 2.2.2. Deceleration

Athletes used the same start protocol and radar technology for the horizontal sprint test and sprinted maximally over 30 m before performing a maximal horizontal deceleration. As previously described [25], we defined the beginning of the deceleration phase as the time point immediately following the maximum velocity achieved during the 30 m sprint. The end of the deceleration phase was established as the lowest velocity following maximum velocity. We used early deceleration phases for analysis using the time point associated with 50% maximum velocity.

#### 2.2.3. Agility Test

Athletes performed the change of direction ability (CODA) test to assess agility. The test consists of performing a forward sprint of 10 m and two lateral runs of 8 m, with the final line of 10 m as a reference. Finally, the athlete returns to the starting line to complete the test (10-8-8-8-10 = 36 m) [33]. A Witty time gate (Microgate, Bolzano, Italy) placed on a tripod at a height of 1 m was used to determine the time. To prevent athletes from activating gates with their arms, they were positioned 50 cm behind the light gate. Each athlete performed two attempts, separated by 4 min of rest. The best record was used for the analysis.

#### 2.2.4. Horizontal Force Velocity Profile

The speed–time data obtained by radar were loaded into the Excel^®^ spreadsheet created by Morin and Samozino [34]. The spreadsheet calculates the maximum horizontal force used during the sprint (F0), maximum velocity (V0), and the maximum horizontal power output (Pmax). In addition, we obtained the proportion of the total force produced by the lower limbs on the floor that is applied horizontally (RFmax) and the rate of decrease in horizontal force as speed increases (D_RF_). This method uses the fundamental laws of movement to obtain the force–velocity relationship using the athlete’s speed and body mass [32]. The use of radar for these purposes was validated using force platforms (absolute bias 3–7%) [17]. Regarding reliability, the mean typical error is small (CV ≤ 8.4%) for all kinetic and kinematic variables [35], making it a valuable field tool for determining these variables. Briefly, the net horizontal anteroposterior GRF ($F_{H}$) applied to the body center of mass (CM) can be modeled over time as follows [17]:$$F_{H}\left(t\right)=m\cdot a_{H}\left(t\right)+F_{aero}\left(t\right)$$
where
m
is the runner’s body mass (in kg) and
$F_{aero}\left(t\right)$
is the aerodynamic drag that must be overcome in a sprint, which is proportional to the square of the velocity of air relative to the runner:
$$F_{aero}\left(t\right)=k\cdot {\left(v_{H}\left(t\right)-v_{w}\right)}^{2}$$
where
$v_{w}$
is the wind velocity (if any) and
k
is the runner’s aerodynamic friction coefficient.

Regarding vertical direction, during the acceleration phase, the runner’s body CM goes up from the starting position to the upright running position and does not change from one complete step to another. Therefore, using the fundamental laws of dynamics in the vertical direction, the mean net vertical ground reaction forces ($F_{V}$) applied to the body CM over each complete step can be modeled over time as being equal to body weight:
$$F_{V}\left(t\right)=m\cdot g\;$$
where g is the gravitational acceleration (9.81 m/s^2^).

The mechanical effectiveness of force application during running could be quantified over each support phase or step by the ratio (RF in %) of $F_{H}$ to the corresponding total resultant ground reaction forces ($F_{Res}$, in N) and the entire acceleration phase by the slope of the linear decrease in RF when velocity increases ($D_{RF}$, in %/s/m):$$RF=\frac{F_{H}}{F_{Res}}\cdot 100=\frac{F_{H}}{\sqrt{F_{H}{}^{2}+F_{V}{}^{2}}}\cdot 100$$

Because the starting block phase (push-off and following aerial time) lasts between 0.5 and 0.6 s [28,29], occurring for an average time of ∼0.3 s, RF and DRF can be computed from $F_{H}$ and $F_{V}$ values modeled for t > 0.3 s.

#### 2.2.5. Statistical Analysis

The descriptive data are presented as mean and standard deviation (mean ± SD). Minimum and maximum (median, quartile 25 (Q_25_) and 75 (Q_75_)) values for each variable are also reported for better interpretation of the data. The study utilized a machine learning approach to examine the relationships between our target variable “x” and a set of predictor variables “y”. Linear regression is a statistical technique that predicts the outcome of a response variable using several explanatory variables and is used to model the linear relationship between explanatory variables and response variables. The model assumes the absence of multicollinearity, which means that the explanatory variables are not highly correlated. This study aimed to determine which independent variables most significantly explain the variation in our target variable. Twelve variables were used to build the models explaining the result of sprint acceleration and agility.

The study’s target (i.e., dependent) variable was time at a different distance in the FVP and the CODA test. All other columns in the dataset were treated as independent variables. In order to avoid multicollinearity, which can bias the interpretation of regression coefficients, a correlation matrix of the independent variables was calculated. In general, if the absolute value of the Pearson correlation coefficient is >0.8, collinearity is likely to exist [36,37]. Thus, variables with a correlation coefficient higher than 0.80 with another variable were removed.

The analysis employed fivefold cross-validation [38,39] to mitigate overfitting and bias, especially given the challenges posed by small datasets. Briefly, fivefold cross-validation divides the dataset into five parts, or “folds”. For each iteration, the model is trained using four folds and validated on the remaining fold. This process is repeated five times, ensuring each fold is the validation set once. To guarantee that all features equally influenced the models, we standardized them to have a mean of 0 and a standard deviation of 1 through standard scaling. Such a step is crucial as variables with different scales can skew the model fit, potentially leading to biased coefficient estimations.

The study adopted a comprehensive approach to model fitting [40] using three linear regression models: Ordinary Least Squares (OLS), Ridge Regression, and Lasso Regression. The OLS regression attempted to minimize the sum of the squared residuals. This model provided an initial understanding of the relationship between the predictor and target variables without penalty imposed on the coefficients. Ridge and Lasso’s regressions introduced a level of bias into the coefficient estimates to manage multicollinearity and improve model interpretability. Ridge Regression includes an L2 penalty that shrinks the coefficients of correlated predictors. Lasso Regression utilizes an L1 penalty that can shrink some coefficients to zero, thus performing feature selection.

A parameter alpha controls the degree of bias or regularization in Ridge and Lasso regressions. Higher alpha values increase the penalty term and thus shrink the coefficients towards zero, effectively simplifying the model. Conversely, an alpha of zero resembles the OLS regression. Therefore, choosing the correct alpha value is critical to balance the model’s complexity and predictive power. The study used a range of potential alpha values to choose the optimal alpha: [0.0001, 0.001, 0.01, 0.1, 1, 10]. The analysis employed RidgeCV and LassoCV, which use cross-validation to select the best alpha that gives the best predictive performance on unseen data [41].

Once the models were fitted with the training data, their performance was determined using the coefficient of determination (R^2^), describing the proportion of the dependent variable’s variance explained by the independent variables. The best explanatory model selection was primarily based on the R^2^ from the test dataset, which reflects the model’s performance on unseen data. Following the selection of the best explanatory model, the model’s performance and stability of the coefficients were evaluated. This involved the calculation of Mean Absolute Error (MAE), Mean Squared Error (MSE), and Root Mean Squared Error (RMSE). These metrics provided a quantitative measure of how our model can explain the actual values [41].

The predicted versus actual values for the training and test datasets were plotted to facilitate interpretation. An identity line was also included to represent perfect predictions, providing a clear visual guide to interpret the model’s performance. To ensure the robustness of our linear regression model, we rigorously examined its underlying assumptions. The normality of residuals was assessed using the Shapiro–Wilk Test, complemented by a QQ plot for visual verification of data conformity to a normal distribution. Furthermore, we verified the homogeneity of variance across levels of the independent variables, employing a visual check via a scatter plot of predicted values against residuals and a statistical assessment using the Breusch–Pagan test. Statistical significance was set at *p* < 0.05. Data analysis was performed in Python 3.9 programming language using the packages “pandas”, “numpy”, “matplotlib”, “sklearn”, “statsmodels”, and “scipy.stats”.

## 3. Results

### 3.1. Descriptive Values

The descriptive analysis of variables associated with the sport contributes to updating the knowledge used to monitor the training process [1]. In addition, the data could be used to compare athletes’ results with international values or similar categories (benchmarking).

#### 3.1.1. Sprint Variables

To determine split times using radar technology for 0–5 m, 0–15 m, and 0–30 m, athletes performed two 30 m maximal sprints. Table 2 shows the times at each distance for the best trial. The times were 1.60 s ± 0.02 s for 0–5 m, 3.21 s ± 0.14 s for 0–15 m, and 5.30 s ± 0.21 s for 0–30 m.

#### 3.1.2. Deceleration and Change of Direction

To describe early deceleration using radar technology, athletes sprint maximally over 30 m, followed by a maximal horizontal deceleration (Table 2). Average values for early deceleration were −3.37 ± 0.87 m/s^2^. The time employed during the CODA test was recorded using time gates to assess agility. The mean time was 9.28 ± 0.30 s.

#### 3.1.3. Mechanical Variables

The speed–time data obtained by radar technology were loaded into an Excel^®^ spreadsheet to determine the mechanical outputs from the best trial of the two 30 m maximal sprints. Table 2 shows the quartile values of the variables from the FVP at 30 m. The mean ± SD values were 5.72 ± 0.49 N/kg for relative F0, 7.93 ± 0.47 m/s for V0, 11.35 ± 1.21 W/kg for relative Pmax, −0.72 ± 0.07 N/s/m for FV slope, 43.2 ± 2.0% for RFmax, and −6.75 ± 0.13 %/s/m for D_RF_.

### 3.2. Mechanical Variables Determining Sprint and Change of Direction Performance

A machine learning approach with three linear regression models was used to determine the best explanatory model for the time at different distances in the FVP and the CODA test. The best explanatory model selection was primarily based on the R^2^ from the test dataset, reflecting the model’s performance on unseen data.

#### 3.2.1. Five Meter Sprint Performance

In many team sports, sprinting occurs over short distances and initial acceleration (0–10 m) is critical to performance [42]. The Lasso regression was the best model for explaining the 5 m time by the mechanical variables relative F0 and Vmax (R^2^ = 0.72; MAE: 0.025; MSE: 0.001; RMSE: 0.033; intercept: 2.8204; relative F0 coefficient: −0.1493; Vmax coefficient: −0.0481; Figure 1).

#### 3.2.2. Fifteen Meter Sprint Performance

The mechanical variables relative F0 and Vmax determine the time taken to cover a distance of 15 m. The Lasso regression model best explains the 15 m time (R^2^ = 0.82; MAE: 0.028; MSE: 0.001; RMSE: 0.038; intercept: 5.6568; relative F0 coefficient: −0.2172; Vmax coefficient: −0.1611; absolute F0 coefficient: 1.5058; Figure 2).

#### 3.2.3. Thirty Meter Sprint Performance

The average sprint distance in field hockey players is less than 20 m [43]. However, we consider it relevant to incorporate the 30 m test for occasions where the athlete must cover longer distances and to determine the FVP profile. Similar to the 5 m and 15 m times, the variables relative F0 and Vmax explained the time taken to cover a distance of 30 m. The Lasso regression model was the best explanatory model for the dependent variable (R^2^ = 0.93; MAE: 0.027; MSE: 0.001; RMSE: 0.035; intercept: 9.6188; relative F0 coefficient: −0.2514; Vmax coefficient: −0.3827; Figure 3).

#### 3.2.4. Change of Direction

Change of direction performance, assessed through the CODA test, is not significantly explained by the FV profile variables and deceleration (R^2^ = 0.04).

## 4. Discussion

The main findings of this study are separated into two components: (i) the descriptive values of a sprint, change of direction, and deacceleration, and (ii) the mechanical variables that determine sprint acceleration performance over short distances and change of direction in elite female field hockey players.

### 4.1. Descriptive Values

According to our first aim, our results show that the time to run 5 m is longer than that of elite team sport female players (0.99 s vs. 1.60 s) [5]. Regarding 30 m, elite female field hockey players cover the distance in 5 s [44], 0.30 s less than the mean of the athletes presented in our study. In addition, a recent study [45] shows values for the times used to run 5, 15, and 30 m in sports similar to field hockey, including female soccer. The times in the three distances are lower than those in our sample (5 m: 1.52 vs. 1.60, 15 m: 3.09 vs. 3.21, and 30 m: 5.16 vs. 5.30 s). These descriptive values allow for categorization of athletes and defining training strategies to improve performance at the international level.

Little evidence exists about the horizontal FVP in female field hockey. A recent study [46] reported the FV profile values of 31 field hockey players (15 males and 16 females). The average values obtained were F0 = 6.88 N/kg, V0 = 7.69 m/s^2^, and Pmax = 13.19 W/kg. Compared with our results, these values are 20.3% higher for F0, 3.0% lower for V0, and 16.2% higher for Pmax. However, direct comparison is challenging due to including male and female participants in their study. Another study describes the values of Norwegian elite athletes from different team sports, such as handball and women’s soccer [47]. F0, V0, and Pmax variables are significantly higher than our sample’s results (~25.4% higher). Conversely, another study [45] showed that soccer and basketball players present Pmax values of 12.6 and 11.4 (W/kg), similar to those measured in our study. These results highlight the need for further research to establish reference values that allow practitioners to guide training and monitor key performance variables in field hockey.

To the best of the authors’ knowledge, this study is the first to show CODA test values in elite field hockey players. Accordingly, there are no specific reference values for female players. This test was initially designed to assess professional football referees’ change of direction abilities [48]. FIFA established a reference value of 10 s for referees in the international category. However, our research observed that the average time is lower than the above reference value, with an average time of 9.28 s. Thus, our preliminary results suggest that athletes demonstrate agility performance that exceeds the benchmark value established for international-level soccer referees. Soccer referees perform actions similar to those in field hockey, such as running, sprinting, side-stepping, and running backward [49]. However, the physical space, top speed, and number and magnitude of accelerations and decelerations occurring in field hockey differ from those of soccer referees [12]. Accordingly, further investigations and establishment of specific reference values for women’s field hockey are necessary to comprehensively evaluate agility in this sport.

Regarding early deceleration, only a few studies present values in team sports such as soccer, rugby, and netball [25]. Those values are higher than those in our study (−3.83 vs. −3.37 m/s^2^). A similar study in team sports showed values of −3.92 m/s^2^ [50], suggesting that the female hockey athletes of our study exhibit lower deceleration rates. However, these values represent mixed values for men and women. Therefore, the differences could be due to the gender of the athletes (mainly males), and not to deceleration performance per se. More information on deceleration in field hockey is needed to determine reference values that can be used as a benchmark for coaches and trainers. Finally, training strategies must enhance athletes’ ability to dissipate braking loads, improving the muscle functions as a shock absorber (energy attenuation) [51]. This approach will lead to new developments in injury mitigation and physical development strategies in team sports [52].

### 4.2. Mechanical Variables

The most relevant mechanical variables determining the times for all the distances studied are relative F0 and Vmax, which partially agree with the hypotheses of this study. Our findings align with Hicks et al. [46], who reported that the same mechanical variables explain 94% of the time taken to cover 30 m. Other studies have also investigated mechanical variables impacting sprint performance over various distances, with speed and applied force emerging as crucial factors [53,54]. The logical relationship between applied force and sprint velocity is underpinned by horizontal velocity’s dependency on ground reaction forces (GRFs) applied in minimal time [55]. The horizontal component of GRF (propulsive impulse) accounts for 57% of the variation in maximum sprint velocity [56]. Our results corroborate the importance of strength training, mainly horizontally applied force, for enhancing sprint capabilities and the need for shorter contact times. Thus, the specific application of force emerges as a decisive factor in sprint performance, surpassing the magnitude of force applied [57]. This challenges field hockey coaches and practitioners, as players must execute maximum-speed sprints while wielding a stick.

The ability to change direction depends on various mechanical variables such as ground reaction force, impulse, velocity, and braking forces [7,58,59]. However, our results showed that mechanical variables derived from FVP and horizontal deceleration do not significantly explain the variance of performance in the CODA test. Our results do not agree with our hypothesis and align with similar results that found low relationships between mechanical variables and female soccer players’ ability to change direction [60]. The FVP profile only identifies horizontally applied force during maximum speed rather than specific braking forces (i.e., horizontal and vertical forces) relevant during COD. Furthermore, the ability to decelerate also depends on variables such as muscle strength (concentric, eccentric, isometric) and technical components [61], which are aspects not addressed in our study.

A limitation of our study is that the frequency sampling of the Stalker radar is 46.9 Hz; however, the literature suggests a frequency between 50 to 250 Hz should be used in motion capture systems [62]. In addition, estimates of variables related to the FVP have been criticized, especially those related to power output. Treating scalar quantities (e.g., power) as a vector could be inappropriate in biomechanics, and vector quantities as impulses could be counted as causative factors in performance [63].

Obtaining the data in a competitive period is a strength of our study, allowing the construction of a valid performance profile. Nevertheless, we recommend assessing the mechanical variables in different general and competitive training periods because training status, training characteristics, responsiveness to training, and nutrition, among other factors, could modify these variables [64]. Finally, control variables such as hormonal profile and anthropometrics could be related to mechanical changes in the training load.

## 5. Conclusions

Our findings highlight the crucial role of understanding female hockey players’ mechanical characteristics of sprinting and COD abilities. By identifying these parameters, researchers and practitioners gain a powerful tool for planning, monitoring, or adjusting training programs. More than just a set of performance metrics, this information provides a comprehensive athlete profile. Such a profile paves the way for individualized training regimes tailored to enhance specific mechanical variables, distances, and skills that need improvement. By refining these individualized training loads, practitioners can support decision making in dynamic team sports. This optimizes performance and plays a crucial role in injury prevention. Consequently, this study is key in setting a benchmark for female hockey training practices, emphasizing a data-driven approach to enhance athletic performance and safety.

## Figures and Tables

**Figure 1 sensors-23-07663-f001:**
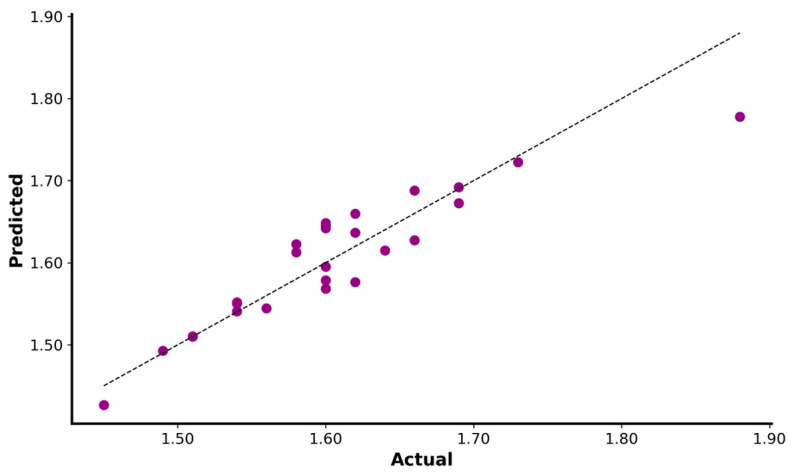
Regression plot of actual versus predicted for 5 m sprint. The black dotted line represents the linear regression function.

**Figure 2 sensors-23-07663-f002:**
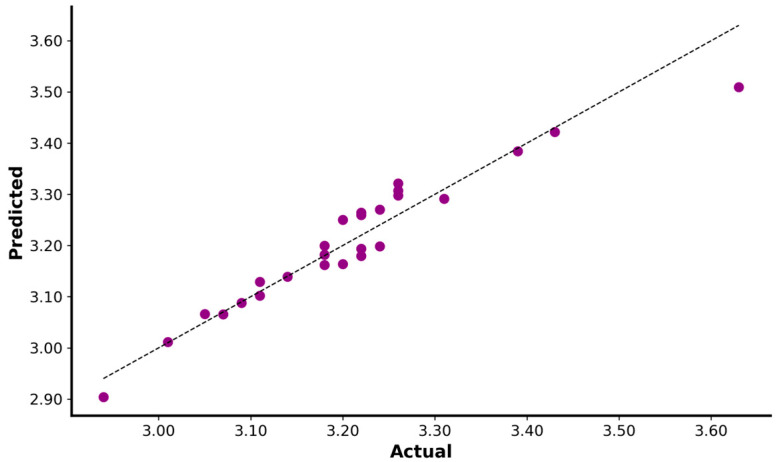
Regression plot of actual versus predicted for 15 m sprint. The black dotted line represents the linear regression function.

**Figure 3 sensors-23-07663-f003:**
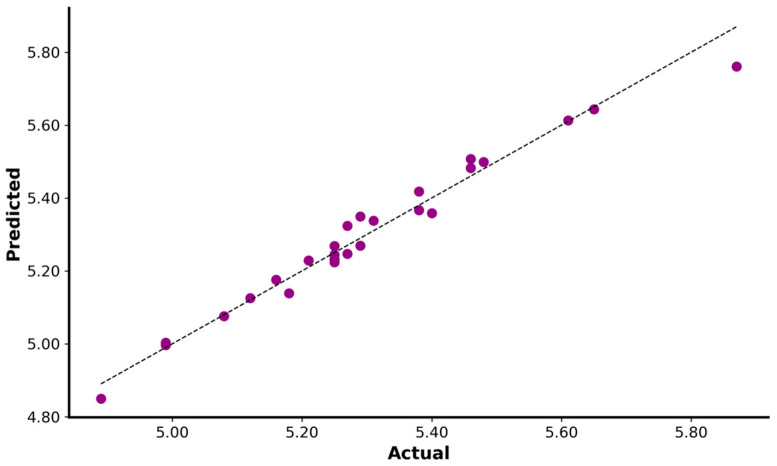
Regression plot of actual versus predicted for 30 m sprint. The black dotted line represents the linear regression function.

**Table 1 sensors-23-07663-t001:** Description of the subjects (mean ± standard deviation).

	Body Mass (kg)	Height (cm)	Age (Years Old)
Defenders (8)	64.8 ± 6.8	166.1 ± 2.1	27.0 ± 3.7
Midfields (11)	62.3 ± 5.0	166.2 ±4.5	24.8 ± 4.6
Forwards (7)	61.4 ± 2.7	166.4 ± 4.3	24.7 ± 4.3
Total (26)	63.1 ± 5.3	166.2 ± 4.1	25.38 ± 4.2

**Table 2 sensors-23-07663-t002:** Descriptive values of times in 5 m, 15 m, and 30 m, mechanical variables of the horizontal profile, times in the agility test, and deceleration (median, quartile 25 (Q_25_) and 75 (Q_75_)).

	Q_25_	Median	Q_75_
5-m (s)	1.55	1.60	1.65
15-m (s)	3.11	3.21	3.26
30-m (s)	5.18	5.27	5.42
F0 (N/kg)	5.34	5.72	6.05
V0 (m/s)	7.5	7.97	8.34
Pmax (W/kg)	10.61	11.51	12.01
FV profile (N/s/m)	−0.79	−0.73	−0.67
RF max (%)	41.28	43.4	44.83
DRF (%/s/m)	−7.33	−6.85	−0.62
Deceleration (m/s^2^)	3.00	3.29	3.69
CODA (s)	9.01	9.23	9.50

F0: maximal theoretical horizontal force; V0: maximal theoretical velocity; Pmax: maximal horizontal power; FVslope: the slope of the linear F-V relationship; DRF: decrease in the ratio of horizontal-to-resultant force; RFmax: maximal ratio of horizontal-to-resultant force. CODA: change of direction ability.

## Data Availability

Data are available for research upon request to the corresponding author.

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
