# Peer review of "Mechanical Determinants of Sprinting and Change of Direction in Elite Female Field Hockey Players"

_sensors, 2023, doi:10.3390/s23187663_

Round 1

Reviewer 1 Report (New Reviewer)

I consider the work very interesting and with a very good statistical analysis. However, I believe that the validity and reliability of the data are too much compromised. On the other hand, from a formal point of view, the presentation of the results and the discussion would need a thorough revision.

Here are some specific considerations:

Line 96: I consider it inappropriate to include goalkeepers in the selected sample to look for a profile of specific field hockey actions that players of this position never perform.

One of the main limitations I observe in this work in the validity of the data obtained with Radar. The authors already speak of a limitation (line 416), related to frequency. In this sense, I consider it essential to demonstrate the validity and reliability of the data obtained.

On the other hand, in the process of data analysis and training of machine learning models, it is essential to have an adequate data set, but also a representative sample of the population under analysis is needed. In this sense, I believe (because there is no data to make me think otherwise) that the sample is too small and therefore the risk of bias too great.

Line 243: It is not appropriate, within the structure of a scientific article, to make evaluations of the results in the Descriptive values section. It would be convenient to do so in the discussion.

The descriptive analysis of variables associated with the sport contributes to 243 updating the knowledge used to monitor the training process [1]. Besides, the data could 244 be used to compare athletes' results with international values or similar categories 245 (benchmarking).

Likewise, it is not necessary in the results to explain again the procedure used to obtain the data.

On the other hand, in the text in the presentation of the data, it is not well explained in which part of the figure appears. Sometimes it is written "...times were 1.61s ± 0.10s for 0-5 m (1A)" others, "...V0 (7.86 ± 0.52 m/s; 1G)".

In section 3.1.2, no reference is made to the fact that the data can be seen in Figure 1.

In the limitations, reference is made to the fact that the data are taken in competitive period and I believe this is a strength. Otherwise, a valid performance profile could not be made. Especially in sports like field hockey, with long periods of competition (leagues).

Author Response

Attached are the answers in word file
Regards

Reviewer 2 Report (New Reviewer)

In the following lines, I would like to give you some suggestions and comments on your work.

In general, this research sounds good; however, it has several aspects that must be improved.

- Abstract

The authors need to partially rewrite this section because the second aim is difficult to understand. 

- Introduction

Please provide the hypothesis at the end of the last paragraph.

-Methods

It will be valuable to inform athletes about their backgrounds (i.e., years of experience, injuries, etc.). At the same time, authors should reconsider placing this information (from lines 97 to 103) in another subsection (i.e., testing).

The statistical analysis section is very interesting. However, there are several doubts regarding this. Due to the small sample size, why did the authors decide to split the data (60-40%) and not use k-fold to avoid possible bias?

Have the authors checked whether their models fulfilled the normality and homoscedasticity of their linear regressions?

Please report on the packages used in the analysis.

- Results

Although Figure 1 is under stable, it would be better if the authors put all the data in a table format with quartiles. I honestly think that this change improves the readiness of the descriptive data.

In the same section, the authors should consider whether they can report the intercept and regression model in the results section. This finding could be valuable for future studies.

Conclusion

The conclusions are based on these results. For this reason, I suggest that the authors highlight the importance of their work.

Author Response

Attached are the answers in word file
Regards

Round 2

Reviewer 1 Report (New Reviewer)

I consider that the modifications made adequately respond to the considerations raised by me.

For this reason, I appreciate and value positively the work done by the authors.

This manuscript is a resubmission of an earlier submission. The following is a list of the peer review reports and author responses from that submission.

Round 1

Reviewer 1 Report

The report was aimed to present the measured mechanical parameters in sprinting of elite female field hockey athletes using radar technology and the analysis for these. The first objective was achieved with the valuable data when the subjects of the examination were twenty-one Chilean national female field hockey team participants. This data is valuable for reference, comparison and evaluation of individual or group athletes’ performances. However, the analysis of the measured results was presented unclearly, lack of logical relationship descriptions between the sprinting parameters that produced confused conclusions.

The opinion of the reviewer is that the report must be major revised to address the shortcoming mentioned above before it can be approved.

Specific comments:

The authors need to make clear logical relationship descriptions between the sprinting parameters RFmax, V0, Pmax, F0, FV slope and the time spent in sprinting distances 5 m, 15 m and 30 m as the statements in lines 33-39 and 157-162.

Additionally, the report had some other important presentation errors such as errors in units of the F0, V0, Pmax and FV slope quantities in Table 2; the r and R^2 parameters in Figures 1, 2 and 3 were not defined in the text; there was no consistency in unit of time in lines 193-194.

The authors also need to make clear the conclusions statement in lines 243-245.

Reviewer 2 Report

It is an honor to review this study. The review of this study is as follows.

This study attempted to define mechanical variables when sprinting using radar in field hockey players

Sufficient analysis of previous studies and the necessity of this study are well presented.

However, it is regrettable that there is not enough information about the results of this study. Information should be provided to readers in an easy way to understand this study.

Other details to be revised are as follows.

1. I recommend you to check the English grammar of the title again and write it.

2. It is difficult to understand this study through the abstract because the abstract does not contain explanations of the abbreviations. It is recommended to include an explanation of the abbreviation in the abstract, referring to the paper published in the existing journal of sensor or the regulations of MDPI.

3. The sentence on line 89-90 is missing the " ) " Please check again.

4. It is understandable that there is no IRB due to the nature of this study, but it cannot be passed on simply that it has been agreed by the players. As with lines 258-260, it is recommended to mention the Declaration of Helsinki in the Method section.

5. Descriptions of the abbreviations given in Table 2 should be included in Table 2. This is also the same with the figures.

6. The "," of the values of r and r2 shown in the figures should be changed to "."

Round 2

Reviewer 1 Report

The report was changed significantly, however, the analysis and reasoning were still not explicitly presented.

The organization of the report was not appropriate, subsection lengths were not equal, especially in the section 3, there were too short subsections.

The study and presentation need to be done more thoroughly and rigorously. These works need more time than a revise cycle. Therefore, the report needs to be resubmitted as a new manuscript.